# Indigenous Cultural Safety Training for Applied Health, Social Work, and Education Professionals: A PRISMA Scoping Review

**DOI:** 10.3390/ijerph20065217

**Published:** 2023-03-22

**Authors:** Tammy L. MacLean, Jinfan Rose Qiang, Lynn Henderson, Andrea Bowra, Lisa Howard, Victoria Pringle, Tenzin Butsang, Emma Rice, Erica Di Ruggiero, Angela Mashford-Pringle

**Affiliations:** 1Dalla Lana School of Public Health, University of Toronto, Toronto, ON M5T 3M7, Canada; 2Department of Psychology, University of Toronto Mississauga, Mississauga, ON L5L 1C6, Canada; 3Department of Clinical Studies, University of Guelph, Guelph, ON N1G 2W1, Canada; 4Waakebiness Institute for Indigenous Health, Dalla Lana School of Public Health, University of Toronto, Toronto, ON M5T 3M7, Canada; 5Centre for Global Health, Dalla Lana School of Public Health, University of Toronto, Toronto, ON M5T 3M7, Canada

**Keywords:** cultural safety, cultural competence, Indigenous health, Aboriginal health, health education, nursing, medicine, allied health, anti-racism, meta-synthesis

## Abstract

Anti-Indigenous racism is a widespread social problem in health and education systems in English-speaking colonized countries. Cultural safety training (CST) is often promoted as a key strategy to address this problem, yet little evidence exists on how CST is operationalized and evaluated in health and education systems. This scoping review sought to broadly synthesize the academic literature on how CST programs are developed, implemented, and evaluated in the applied health, social work and education fields in Canada, United States, Australia, and New Zealand. MEDLINE, EMBASE, CINAHL, ERIC, and ASSIA were searched for articles published between 1996 and 2020. The Joanna Briggs Institute’s three-step search strategy and PRISMA extension for scoping reviews were adopted, with 134 articles included. CST programs have grown significantly in the health, social work, and education fields in the last three decades, and they vary significantly in their objectives, modalities, timelines, and how they are evaluated. The involvement of Indigenous peoples in CST programs is common, but their roles are rarely specified. Indigenous groups must be intentionally and meaningfully engaged throughout the entire duration of research and practice. Cultural safety and various related concepts should be careful considered and applied for the relevant context.

## 1. Introduction

There is substantial evidence to suggest that anti-Indigenous racism is a widespread health and social problem in Canada. While the term, Indigenous, refers to a variety of Aboriginal groups around the world [1], in Canada, Indigenous refers to the country’s first inhabitants, namely, First Nations, Inuit, and Métis peoples, as established in Section 35 of the Canadian Constitution [2]. Racism may be understood as “racist ideologies, prejudiced attitudes, discriminatory behaviour, structural arrangements and institutionalized practices resulting in racial inequality as well as the fallacious notion that discriminatory relations between groups are morally and scientifically justifiable” [3]. According to several population-based studies undertaken with Indigenous peoples across Canada, between 39% and 43% of respondents reported experienced racism [4,5,6,7]. Systemic anti-Indigenous racism in health service organizations across the country is equally problematic. The investigations into the deaths of Brian Sinclair in Manitoba in 2008 [8] and Joyce Echaquan in Quebec in 2020 [9], coupled with the alarming findings from the anti-Indigenous racism investigation in British Columbia’s health care system in 2020 [10] collectively demonstrate the insidious nature of anti-Indigenous racism in Canada’s health systems. Not only is racism a serious challenge for Indigenous peoples while accessing care, but also it has profound negative impacts on their likelihood of accessing future care [10,11,12,13,14,15,16,17].

Beyond the health sector, research evidence also points to widespread anti-Indigenous racism in Canada’s social work [18,19] and Kindergarten to Grade 12 education systems [20,21]. Moreover, Indigenous peoples have been identified as the most disadvantaged with respect to accessing education in English speaking, colonized countries such as Canada, United States, Australia, and New Zealand [20].

In 2015, Canada’s Truth and Reconciliation Commission (TRC) recognized the legacy and impact of residential schools on Indigenous peoples across the country [22] and put forth 94 Calls to Action, including for the health, social work/child welfare, and education systems. These calls included providing cultural competency training for health professionals to address unconscious bias and systemic racism and developing culturally appropriate education curricula, while building student capacity for intercultural understanding, empathy, and mutual respect [23]. The calls also set out the need to ensure that social workers who conduct child-welfare investigations be properly educated and trained about the history and impacts of residential schools on children and their caregivers. In terms of education systems, the TRC recommended that resources be provided to ensure Indigenous schools utilize Indigenous knowledge and teaching methods in the classroom and that relevant teacher-training needs be identified to ensure such knowledge and methods are implemented.

Initiatives such as Canada’s TRC, along with similar federal government initiatives in Australia and New Zealand, have led to the emergence of new training programs to facilitate cultural safety and competence and, ultimately, to eliminate anti-Indigenous racism and discrimination in health, social work, and education systems [24,25]. However, given the nascency of the field of Indigenous cultural safety training combined with the need to expand the availability of such training to address anti-Indigenous racism, there is an urgent need to synthesize and understand existing evidence on how such training programs are conceptualized and developed, as well as how they are implemented in practice and evaluated for impact.

## 2. Objectives

Our team sought to synthesize the academic literature that conceptualizes and/or operationalizes Indigenous cultural safety training within the fields of health, social work, and education. The aim of this study was to answer the following three questions: (1) What is the general state of knowledge on Indigenous cultural safety training in the applied fields of health, social work, and education? (2) What methods are used to develop, implement, and evaluate Indigenous cultural safety training for students and professionals in the applied fields of health, social work, and education? (3) What content themes are included in existing Indigenous cultural safety training program for students and professionals in the fields of health, social work, and education?

## 3. Methods

### 3.1. Protocol and Registration

This study involved a scoping review that included the PRISMA extension protocol for scoping reviews (PRISMA-Scr) proposed by Tricco et al. [26]. A detailed research protocol for this scoping review is published in *Social Science Protocols* doi.org/10.7565/ssp.2020.2815 [27], an open-access online journal platform. The authors of this paper chose not to register this study, both because of the rigid methodological requirements for registration, which do not align well with Indigenous research approaches, and because no other Indigenous studies were registered at the time. Indigenous cultural safety is a nascent area of research and practice, and the goal of this review was to achieve a comprehensive search. Our analysis sought to draw out insights that could inform Indigenous cultural safety training for professionals in the applied fields of health, social work, and education, which work with Indigenous peoples in Canada.

### 3.2. Eligibility Criteria

We reviewed the global academic literature on Indigenous cultural safety training that included but was not limited to Indigenous peoples in Canada. Inclusion criteria: This review was restricted to articles about Indigenous cultural safety in health, social work and education in British colonial settler nation states, including Australia, New Zealand, Canada, and the United States. All peer-reviewed primary research articles on the topic of Indigenous cultural safety within the fields of health, social work, and education that were reported in the academic literature between 1996 and 2020 were included. The review dates were selected to ensure that the full history of the term “cultural safety”, first coined in 1996 [28], was captured. This review included all articles published in or translated into English. The exclusion criteria included articles published prior to 1996; articles published in countries other than New Zealand, Australia, United States, or Canada; articles focused on fields other than health, social work, or education; and research published as non-academic literature or in languages other than English.

### 3.3. Information Sources

To identify potentially relevant literature in health, social work, and education, the following five bibliographic databases were searched from 1996 to 2020: Medline, EMBASE, CINAHL, ERIC, and ASSIA. These databases were selected to capture the fields of health, education, and social work as aligned with the focus of this review. The search strategies were developed by an experienced librarian at the university where the authors are affiliated and further refined through team discussion.

### 3.4. Search Strategy

A three-step search strategy [29] was utilized for this review. The first step involved a limited search of two initial databases, Medline and EMBASE, followed by an analysis of subject headings and search terms based on identified titles and abstracts. A second search was then conducted using all identified subject headings and keywords across all five databases (see Table 1: Search Strategy).

### 3.5. Selection of Sources of Evidence

The selection of studies for this review was performed in Covidence independently by two reviewers and involved a four-stage process. The first stage of screening involved identifying and removing duplicate studies from Covidence. The second stage of screening involved reviewing the titles and abstracts of the remaining articles and applying the inclusion and exclusion criteria. Any disagreement between the two reviewers on applying the criteria was resolved through discussion.

### 3.6. Data Charting Process

A data-charting tool was developed for this study with input from several co-authors and a librarian at the university where the authors are affiliated. The data-charting tool was developed in Google Spreadsheet, and several decisions were made to establish the focus, nature, and scope of data to be extracted, including (1) to cut and paste entire “chunks” of text verbatim into the tool rather than paraphrasing the data; (2) to include article page numbers associated with data on cultural safety concepts and critiques; and (3) to include both the study aim, along with the research question if explicitly stated and relevant to the study. The data-charting tool was then piloted by all members of the review team based on two pre-selected articles. Several procedural decisions were made during the pilot stage to further articulate the scope of data to chart, both for existing data points and for two new data points added during this stage. These decisions included (1) whether the paper included Indigenous authors or Elders; (2) identifying whether a data collection or evaluation tool was included or referenced; (3) specifying that “training modality” referred to the format (e.g., in-person, online, etc.) and “training components” related to the topics covered; (4) including only “yes” or “no” on four specific data points; (5) specify that “duration of time the training program has been running” was in relation to the study publication year; (6) adding “recommendations outlined in the article”; and (7) adding “stated limitations of the evaluation”.

The data-charting process involved charting each article twice, and this was undertaken between August 2020 and June 2021. The first author independently completed one round of data-charting, and the second round was completed by a team of six reviewers independently. Several questions arose among the reviewers throughout the data charting process concerning the nature and amount of data to be extracted and charted, and these issues were generally resolved through discussion as a team. Any disagreements between two reviewers were resolved through dialogue or further adjudication by a third reviewer where necessary.

### 3.7. Data Items

The items included in the data-charting tool were developed deductively and addressed four main themes. The first theme included Article Characteristic for Indigenous cultural safety training for the sources of evidence included in the review, namely, bibliographic details; country of study; whether author(s) identified as Indigenous; study aim/objective(s); target field(s)/discipline(s) (i.e., medicine/physicians; nursing/midwifery; allied health; health, general; education; and/or social work); and target population (students and/or professionals). The second theme concerned Cultural Safety Concepts, Critiques and Rationale, which included cultural safety and related concepts either articulated or cited by study authors, along with critiques of the concepts, and the rationale given for why cultural safety is necessary in the fields of health, education, and social work.

The third theme involved the Characteristics of Cultural Safety Training described in the sources of evidence. Data themes included who sponsored and developed the training program, whether Indigenous scholars and practitioners or knowledge keepers were involved in the development process, and the roles of community partners who were engaged. Other themes included who delivered and received the training program, along with details of the training program itself—such as the objectives, delivery modality, component descriptions, delivery duration and timeline, details of any post-program support, and the programs’ stated limitations. The fourth and final theme addressed Evaluation Details of Cultural Safety Training. These data involved the evaluation objectives, data collection methods, type(s) of data collected (i.e., qual, quant, and both), evaluation results, stated evaluation limitations, recommendations for future cultural safety training programs, and whether a data collection tool was provided (see Appendix A).

### 3.8. Synthesis of Results

Once the data were charted, the data synthesis process was undertaken in three stages and carried out by three reviewers. The first stage involved analyzing both extractions of data for each theme to identify and resolve any discrepancies. All inconsistencies that were discovered were resolved through discussion and/or revisiting the sources of data where needed. The second stage involved analyzing data on the general characteristics of Indigenous cultural safety within the studies selected for the review, a process which involved categorizing the data within each data theme and then counting the frequency of findings within each category. The third stage involved synthesizing data that described a cultural safety training and/or the evaluation of a cultural safety training. The last stage involved synthesizing the data within each theme to create an overview of the data characteristics of Indigenous cultural safety, then categorizing the data within each theme into distinct groups and counting the frequency of findings within each category. Finally, the process of data synthesis also involved excluding or refocusing some of the data themes that were introduced in the protocol paper and proposed as part of this study. The reasons for these changes were varied and are set out in Section 4.4: Details of Indigenous Cultural Safety Training Intervention Evaluations.

## 4. Results

### 4.1. Selection of Sources of Evidence

The database searches were conducted in May 2020 and resulted in a total of 3600 sources, which were imported into Covidence online screening software (see Figure 1: Prisma Flow Diagram). Covidence detected and removed 1117 duplicate articles and 2153 irrelevant studies. In the end, 320 full-text articles were screened and assessed for eligibility within Covidence by two reviewers. Through this process, an additional 178 were excluded for the following reasons: (a) not related to cultural safety training and/or cultural safety conceptualization (*n* = 126); (b) not peer reviewed (*n* = 36); (c) conference summary (*n* = 8); (d) duplicate content (*n* = 2); (e) literature review (*n* = 2); (f) not focused on our target professional groups (*n* = 2); (g) full text of article not found/available (*n* = 1); and (h) published before 1996. While 142 remaining studies were selected for inclusion, an additional eight articles were removed during the process of data charting because either the authors and content of these articles overlapped significantly with other selected articles (*n* = 3), or the paper involved Indigenous peoples’ review of a cultural safety training but did not focus on the targeted professionals or students and their experiences with the training (*n* = 5). Data from a total of 134 included full-text articles, including quantitative and/or qualitative research studies as well as expert opinion pieces, were extracted for this study (for a complete list of studies selected, see Appendix B).

### 4.2. General Characteristics of Sources of Evidence

Table 2 presents the general characteristics of the 134 sources of evidence selected for this review. The citations for each finding presented in this table are included in Appendix C.

### 4.3. Details of Indigenous Cultural Safety Training Interventions

Of the papers that described an Indigenous cultural safety training program (*n* = 69), more than two-thirds mentioned the involvement of Indigenous peoples in the training development process (*n* = 48; 70%) (see citations for these finding in Appendix D). A wide variety of education approaches to delivering cultural safety content were illustrated as part of these interventions, with all having described more than one modality. Teaching/lectures was the most common training modality, described in almost a third of papers (*n* = 21; 30%). Workshops (*n* = 18; 26%), discussions sessions (*n* = 16; 23%), and immersive experiences/community visits (*n* = 13; 19%) were also common, detailed by an estimated quarter to approximately a fifth of relevant articles.

Following these modalities, storytelling/yarning reflective exercise/practice online content, and video/videocasts were each described by an eighth of chosen studies (*n* = 8; 12%). Also common among training modalities were clinical placements, case-studies, tutorials, and workbooks/readings, each of which were included as part of a tenth of interventions (*n* = 7; 10%). Finally, Indigenous Elders/mentors/community supports (*n* = 6; 4%), course-based modules (*n* = 5; 4%), and group work (*n* = 4; 3%) were each described by a few papers. The least common activities mentioned were drama/role-play exercises, and a visit to a gallery/museum, each mentioned by a couple of articles (*n* = 2; 3%). Notably, eight papers (12%) described the provision of follow up support for learners of cultural safety beyond the primary training period.

With respect to the training timeline, the largest proportion of articles detailing a cultural safety training intervention described the training as lasting 1–5 days/sessions (*n* = 15; 22%). Following this, in approximately one-sixth of selected articles, the training involved part of a one-semester undergraduate course (8–25 h) (*n* = 12; 17%). The shortest training involved a one-off lecture/workshop/cultural visit, which was mentioned in an estimated one-sixth of articles (*n* = 10; 15%), followed by a 1–5 week immersive experience in one of ten articles (*n* = 7; 10%). The longest training experiences involved embedded content across a four-year bachelor’s degree (unknown total hours) followed by training of 6–10 days/sessions, with each of these approaches occurring in one of ten articles (*n* = 6; 9%). Three training durations, namely, 3–12 months of immersion, a one-semester undergraduate course (42 h), and a partial undergraduate course over two semesters (50–52 h), each occurred in an estimated one-fifth of papers (*n* = 14; 20%). The two least frequent training timelines were included in one article each (*n* = 1; 1%), namely, an intensive course over two semesters (148 h), and two years of content embedded in a health service organization. Nearly one in five articles did not include details on the duration of cultural safety training (*n* = 13; 19%).

In terms of delivering cultural safety training interventions, university professors/administrators were mentioned in approximately twenty-five articles that described cultural safety training (25; 36%). Clinicians/clinical instructors (health professions) (*n* = 20; 30%), and Indigenous Elders/mentors/cultural educators/community/organizations (*n* = 19; 28%) were each mentioned in approximately one-third of relevant articles. University/college lecturers/educators delivered the training in an estimated one-fifth of articles (*n* = 15; 22%), followed by researchers in approximately one in twenty-five articles (*n* = 5; 7%). Allied health professionals and consultants were each mentioned in the context of training delivery in only a few articles (*n* = 3; 4%). Two articles provided no details on who delivered the training, while one article described non-clinician health staff as responsible for training delivery. Finally, half of the articles that described cultural safety training reported that Indigenous people(s) were involved in delivering the intervention (*n* = 35; 51%). Indigenous peoples involved in delivering cultural safety training includes both those categorized as “Indigenous Elder/Mentor/Cultural Educator/Community/Community organization” as well as those in other relevant categories listed immediately above.

### 4.4. Details of Indigenous Cultural Safety Training Intervention Evaluations

Nearly one-third of papers selected for this review described an evaluation of an Indigenous cultural safety training intervention (*n* = 61; 46%). The citations of these finding are included in Appendix E. These particular studies involved a broad range of objectives, with the largest proportion of papers (*n* = 19; 31%) aimed broadly at understanding leaners’ general experiences, perceptions, needs, and/or preferences related to a cultural safety training in which they participated. Another paper similarly focused on understanding perceptions of a given training intervention but from the perspective of Indigenous community members rather than participants (*n* = 1; 2%).

Other common evaluation approaches involved exploring participants’ reports of learning experiences, outcomes, and/or barriers as intervention outcomes (*n* = 16; 26%), and the new skills, behaviours, and/or practices that learners acquired (*n* = 15; 25%). Similarly common, reported in nearly one-fifth of relevant papers, involved understanding learners’ new knowledge, awareness, attitudes, and confidence since the intervention (*n* = 11; 18%). Other evaluation approaches included learners’ perceptions about Indigenous peoples and/or their health issues, as well as learners’ receptivity and/or resistance to intervention content, each mentioned in one intervention evaluation (*n* = 1; 2%). A less common approach to evaluating cultural safety training interventions among the selected articles involved exploring processes, and specifically intervention planning and implementation processes (*n* = 2; 3%), and leaners’ behaviour change processes (*n* = 2; 3%). Another paper looked at the relationship between learners’ intervention participation (intervention exposure) and their perceptions of Indigenous peoples (the outcome). Finally, several papers considered the context wherein the training intervention was implemented, specifically, the organizational context as an enabling factor (*n* = 5), the impact of the training on organizational processes and priorities (*n* = 3), and new organizational policies, positions, capacity, and/or mandate related to the indigenous cultural safety training (*n* = 2). In terms of the methodological approaches adopted to evaluate cultural safety training interventions, just over half of relevant articles described utilizing both qualitative and quantitative methodologies (*n* = 32; 52%). Another third of articles outlined the use of qualitative methods only (*n* = 20; 33%), while approximately one in six papers used quantitative methods only as part of their intervention evaluations (*n* = 9; 15%). A broad range of research methods was taken up to evaluate the influence and/or impact of cultural safety training interventions on participants. Data collection methods prior to the intervention involved pre-surveys (*n* = 13; 21%), only four of which included open-ended, qualitative questions (*n* = 6; 8%). Less common were pre-focus groups (*n* = 2; 3%) and a pre-test (*n* = 1; 2%). A small number of evaluation papers undertook observations/reflections half way through the interventions (*n* = 4; 6%).

The greatest range of data collection methods occurred immediately following the intervention, with post-surveys being the most common approach, mentioned by two-thirds of relevant articles. A large proportion of these post-surveys focused only on quantitative data (*n* = 24; 39%), while others also included open-ended qualitative questions (*n* = 12; 20%) Participant interviews were also common following an intervention, also mentioned in a fifth of relevant articles (*n* = 12; 20%). Oral or written learner feedback (*n* = 6; 10%), and post-focus groups (*n* = 6; 10%) were each mentioned in a tenth of papers describing a training evaluation, followed by learner reflections or case studies (*n* = 4; 6%) and learner journal entries or digital storytelling (*n* = 3; 5%), as described in a quarter of relevant papers. A post-test, talking circles, analysis of developed curriculum, and post-researcher observations/reflections were each mentioned in only one relevant study (*n* = 1; 2%). Finally, a small number of relevant studies involved data collection after a period of delay following a cultural safety training intervention. These methods included a delayed post-survey undertaken at various stages between 3 and 55 months following the intervention (*n* = 2; 3%), while a third paper described delayed learner reflections collected at 1–2 weeks following the intervention (*n* = 1; 2%). Finally, of the papers that described a cultural safety training intervention, only approximately half of these papers presented the limitations of the methods chosen for the intervention evaluations (*n* = 33; 54%).

The review team chose to exclude or refocus some of the data themes that were introduced in the protocol paper and proposed as part of this review study. The reasons for these changes were varied. First, data related to some of the data themes were largely homogenous. These themes included, for example, the rationale for providing training on cultural safety (or related) concepts—reasons which overwhelmingly involved reducing disparities between Indigenous people and relevant settler populations on health outcomes (for health-related papers) or education achievement (for education related papers). Data were also not presented in this paper on themes wherein the data were not described consistently across selected studies. This included themes such as description of training components, number of people who completed the training, and details of post-training supports provided. Similarly, data on who sponsored or developed the training intervention, as well as the duration of the training intervention and whether it was implemented in rural or urban settings, were largely incomplete, with a relatively small proportion of articles including fulsome descriptions for these themes. For its part, data on “the aim/objective” of the evaluation tended to be very general, including reasons such as “to evaluate a cultural safety training intervention”, which was therefore replaced with “evaluation focus”, which instead centred on the nature of data collected (i.e., participants perceptions, intervention processes or outcomes, etc.). Finally, the evaluation objectives and recommendation were not analyzed and included in this review because these data were rich in description, and synthesizing the findings would require undertaking content analysis, which the authors determined was beyond the scope of this review given the timeframe required to complete the study.

## 5. Discussion

### 5.1. Summary of Evidence

This study sought to synthesize the state of knowledge on Indigenous cultural safety training in the applied fields of health, social work, and education. This paper sets out the broad range of approaches used to develop, implement, and evaluate these training programs, along with the substantive themes included in the trainings.

This study found a significant growth in research about Indigenous cultural safety training in the three broad fields over the last few decades, and primarily within the last ten years. Half of the papers selected described cultural safety training, with the greatest share of these articles targeting the field of nursing/midwifery followed by medicine/physicians. This finding is unsurprising considering that nurses/midwives tend to comprise the largest share of the health service workforce, followed by physicians [30]. The education field, for its part, comprised only a tenth of selected articles, suggesting that much more is needed to improve access to cultural safety training for educators. Across the four broad fields, approximately half of the population targeted by the articles were students while the other half were professionals. This is an important finding as it suggests that not only are students learning about cultural safety within their post-secondary education programs, but so are instructors, mentors, and employers working in health, social work, and education fields. This may indicate that professionals have the capacity to contribute to enabling environments for new graduates to practice cultural safety when they begin their careers.

A substantial variety of training modalities was apparent in the training interventions, with teaching or lectures, workshops, and immersive experiences or community visits among the most common. Similarly, there was a huge range in the timeline or duration of cultural safety trainings. Interventions varied from several individual sessions to one or more months of immersive experience, to fully embedded content within a post-secondary course or throughout an entire undergraduate degree program. Even the actors who delivered the training ranged significantly, from university professors and health service clinicians, to lecturers, researchers, and consultants.

There was little consistency in the objectives and methods to evaluate Indigenous cultural safety training interventions. Some evaluations focused on the experiences of learners and how they made sense of those experiences, while others looked at specific outcomes or processes (or the relationships between these two factors). There were also evaluations that considered the contexts within which the interventions were implemented and how these contexts influenced the interventions. Methodological approaches chosen for intervention evaluations also varied widely, including qualitative, quantitative, and mixed methods approaches. Most studies collected data after a cultural safety intervention, while fewer considered relevant data prior to, during, or following a delay after an intervention. Pre- and post-surveys were the most common method adopted (with some including open-ended questions), followed by interviews, focus-groups, and methods involving written reflections, case studies, journal entries, and/or storytelling. Notably, only half of the studies included set out the limitations of the methods chosen for intervention evaluations.

Only a fifth of papers selected for this review included a named Indigenous author, while an estimated one-tenth of papers failed to specify even the broad Indigenous population relevant to the study (e.g., First Nations, as in Canada). Moreover, while a substantial proportion of these training interventions involved Indigenous people(s) in the development processes, the specific roles they undertook were rarely mentioned. Similarly, approximately half of the training interventions involved Indigenous Elders, experts, and other community members in the training delivery process, yet how they contributed to the training vis-à-vis other actors involved was not clearly articulated. These findings highlight the need to ensure Indigenous voices are central to identifying and prioritizing the content and approaches to be included in Indigenous cultural safety trainings [31]. Moreover, a greater commitment to meaningfully engage Indigenous communities on projects concerning them is urgently needed by researchers and educators working to advance cultural safety in the health, social work, and education fields [31,32].

Only half of the selected papers described and/or cited a cultural safety or similar concept. This is a somewhat surprising finding given that this review sought to understand the concept of cultural safety and how training interventions have attempted to facilitate its achievement. It is notable that many different concepts and definitions related to cultural safety emerged from the selected studies and were often used interchangeably. These concepts, in part, included cultural competence, cultural sensitivity, cultural responsiveness, cultural knowledge, cultural awareness, and cultural humility. Confusion related to these cultural concepts in selected studies was made worse by several factors, including (1) few attempts to justify why a given concept was chosen over another; (2) original sources for these concepts tended to not be clearly cited; and (3) little demonstrated understanding and contextualization of the literature wherein these concepts emerged. Similarly, and perhaps unsurprisingly given the current nascent state of this literature, less than a fifth of papers provided a critique of a cultural safety or similar concept.

There is evidence to suggest there is some convergence around the geopolitical origins of two concepts, namely, cultural safety and cultural competence. Cultural safety emerged in New Zealand by a Māori nurse working within the health system at a time when the country was primarily bi-racial, comprising Indigenous peoples and Western European settlers [28]. Cultural competence, for its part, is a term developed in the United States within the transcultural nursing movement and in response to nurses’ lack of understanding of the unique health needs of immigrants [33]. In addition to the convergence of these two concepts, Lauren Baba, a Canadian researcher at the National Collaborating Centre for Indigenous Health in British Columbia, published a 2013 report that draws upon multiple sources to conceptualize and articulate the distinctions between cultural safety, cultural competence, and other similar concepts, including cultural awareness and cultural sensitivity [34]. These four cultural related concepts and their definitions are outlined in Table 3 (Defining Cultural Safety, Cultural Competence, and Similar Concepts).

Both academic research involving these cultural related concepts and training programs that seek to facilitate their development would benefit from considering, comparing, and contrasting the concepts set out by Baba [34]. Not only would consolidating these terms across the literature work to further develop the scholarly work in this area, but also it would facilitate the advancement of the field more broadly by furthering the understanding of which concepts are preferable in different contexts and how to ensure their achievement through targeted training programs. This field of research would also benefit from advancing understanding on how these different concepts relate to one another in practice, how they should be applied, and the merits and drawbacks of each. Moreover, careful attention must be paid to the contexts in which these concepts are applied given the unique socio-political and historical situations in which some of them have emerged [28,33].

The increased attention to cultural safety and related concepts has been fueled by national initiatives to address the historical injustices against Indigenous peoples. For example, Canada’s Truth and Reconciliation Commission’s Calls to Action (2015) set out the need for cultural safety training for healthcare professionals, and federal funding for post-secondary institutions and educators to establish national research programs in collaboration with Indigenous peoples to advance Reconciliation [23]. A similar 2008 initiative in Australia, entitled Closing the Gap [24], called for multi-sectoral action in collaboration with Aboriginal communities to improve Indigenous health, which was followed in 2019 by the development of a National Indigenous Agency in 2019 responsible for leading and coordinating the implementation of the strategy [35]. Australia’s commitment to action was published almost a decade before Canada’s Truth and Reconciliation Commission’s Final Report and Calls to Action, which may explain why an overwhelming majority of studies selected for this review were published in Australia—nearly 2.5 times the number of studies published in Canada, and more than three times the number published in the United States. 

### 5.2. Limitations

The main limitation of this study is that the data presented is somewhat dated. The article searches were undertaken in mid-2020, in the early stages of the COVID-19 pandemic, and with no dedicated research funding. New priorities to understand and address Indigenous peoples’ experiences with the COVID-19 became competing priorities with this study, and at a time when there was increased demand for Indigenous CST. As a result, our team used the preliminary findings from this review to prioritize the development of Oshki M’naadendimowin, a 36 h Indigenous cultural safety training program for faculty members, staff, and students at the Faculties of Medicine, Nursing, Social Work, and Public Health at the University of Toronto. For these reasons, combined with the large number of articles and data categories included in this review, we were not able to update the article searches to 2023. There is also the possibility that some relevant studies may have been missed due to the selection of databases or the exclusion of grey literature. The language skills of the researchers on the team also limited the searches to English publications, where French, Spanish, or Indigenous languages may have been used within the selected geographic regions. Furthermore, search terms for populations did not include Indigenous nation-specific names (e.g., Mohawk, Cree), and thus publications that did not use overarching terms such as Indigenous or First Nations may have been missed.

## 6. Conclusions

This review presents the substantial work undertaken over the last three decades in Canada, Australia, New Zealand, and the United States to deliver Indigenous cultural safety training interventions to address anti-Indigenous racism and its historical legacy in national, publicly funded health and education systems. While the research findings from this review set out the substantive progress toward developing training interventions, this field of research remains in its infancy. Future research on cultural safety and related training interventions requires greater clarity in conceptualization of cultural related concepts, and more robust approaches to evaluate such interventions and identify which training strategies are most effective. Moreover, future work is required to identify how and when cultural safety and related concepts should be applied, and this is likely to vary across countries, regions, and relevant Indigenous groups.

Finally, to advance future research and practice on Indigenous cultural safety, relevant Indigenous groups must be intentionally and meaningfully engaged throughout the entire duration of research and practice. A 2021 report published by the Yellowhead Institute on Canada’s progress toward reconciliation concluded that symbolic actions were being prioritized by the Canadian government over lasting permanent and structural changes necessary to transform the country’s relationship with Indigenous peoples [36]. It is imperative that current momentum to advance the field on Indigenous cultural safety training is capitalized upon to address anti-Indigenous racism and the legacy that colonial governments continue to impose upon Indigenous peoples in counties such as Canada.

## Figures and Tables

**Figure 1 ijerph-20-05217-f001:**
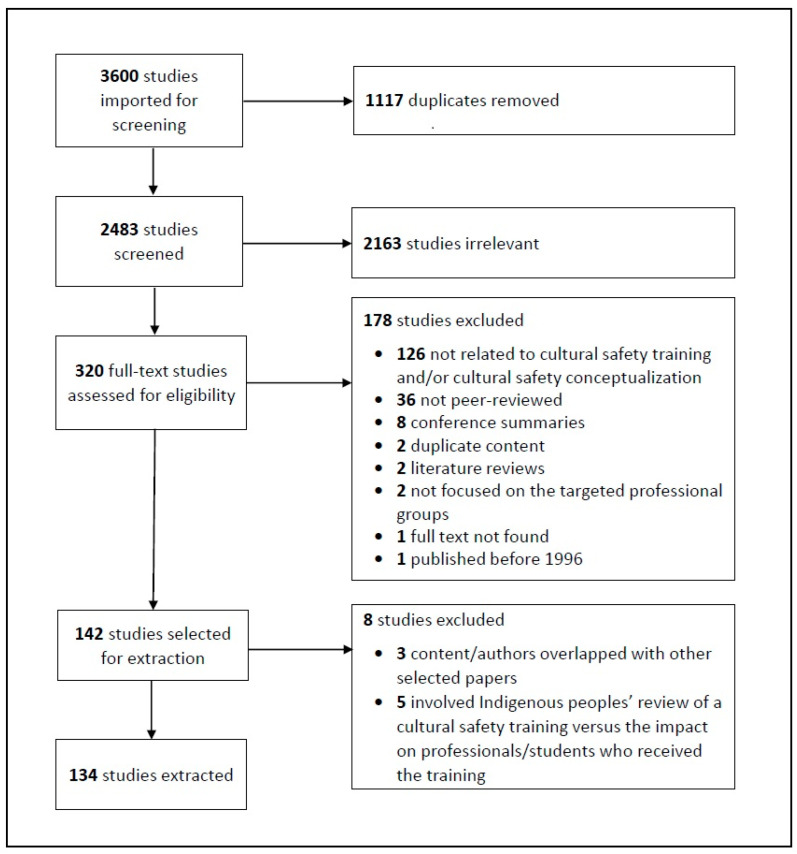
Prisma Flow Diagram.

**Table 1 ijerph-20-05217-t001:** Search Strategy.

Subject Heading Search		
Culturally Competent Care OR Cultural Competency OR Cultural Competence OR Cultural Safety OR Cultural Sensitivity AND Indigenous Peoples OR American Native Continental Ancestry Group OR Oceanic Ancestry Group AND Health Personnel OR Education OR Curriculum OR Teaching OR Social Work OR Social Work Service OR Students, Social Work OR Education, Social Work OR Social Service Assessment OR Facilities, Manpower and Services OR Occupational Health Services		
Keyword Search		
Search categories		Search terms
Activity	Indigenous cultural Safety Training	“cultural safety” OR “cultural competence” OR “culturally appropriate” OR “cultural sensitivity” AND
Context	Indigenous	Indigenous OR “First Nations” OR “Métis” OR “Inuit” OR “Aboriginal” OR “Maori” OR “Torres Straight” AND
Population	Health	“health care” OR “practitioner” OR “health care provider” OR “health professional” OR nurse OR physician OR “public health” AND
	Education	“education” OR “teacher” OR “faculty” OR “curriculum” AND
	Social Services	“social work” OR “child welfare” OR “criminal justice work” OR “support work” OR “employment support” OR “housing service” OR “family service” or “child aid” OR “child service” OR “youth service”
Databases selected		Medline, Embase, CINAHL, ERIC, and ASSIA

**Table 2 ijerph-20-05217-t002:** General Characteristics of Sources of Evidence.

General Characteristics	Number (*n* = 134)	Percent (%)
Publication Year		
2011–2020	93	69%
2001–2010	35	26%
1996–2000	6	5%
Country(ies) of Study *		
Australia	66	49%
Canada	28	21%
United States	21	16%
New Zealand	20	15%
Author(s) Self-identified as Indigenous		
Yes	26	20%
Relevant Indigenous Group Specified in Article		
Aboriginal (Australia)	66	49%
Torres Strait Islander Australia)	46	34%
Māori/Tangata Whenua	20	15%
First Nation (Canada)	18	13%
American Indian (United States)	17	12%
Métis (Canada)	12	9%
Inuit (Canada)	11	8%
Unspecified (Canada)	10	7%
Alaska Native (United States)	6	4%
Kānaka Maoli/Native Hawai’ians (United States)	2	1%
Unspecified (United States)	2	2%
Target Discipline(s)/Field(s)		
Nursing/midwifery	45	34%
General health ^±^	31	23%
Medicine/physicians	25	19%
Allied health ^¥^	18	13%
Social work/services	15	11%
Education ^α^	13	10%
Target Population(s)		
Students only	64	48%
Professionals/academics only	62	46%
Professionals/academics and students	8	6%
Cultural Safety (or similar) Concepts and Critiques		
Described and/or cited a cultural safety (or related) concept	69	51%
Described and/or cited a critique of a cultural safety (or related) concept	26	19%
Cultural Safety Training Interventions and Evaluations of Interventions		
Described an Indigenous cultural safety training Intervention	69	51%
Described an evaluation of an Indigenous cultural safety training intervention	61	46%

* A study undertaken in both Canada and the United States was counted twice. ^±^ General Health refers to Public Health professionals, Psychologists, health researchers, and otherwise unspecified health professionals or practitioners. ^¥^ Allied Health Professionals refers to Dentists, Pharmacists, Occupational Therapists, Physiotherapists, Audiologists and Speech Language Therapists, and Radiologists. ^α^ Education refers both to those in the field of Education and to university students/staff in unspecified fields.

**Table 3 ijerph-20-05217-t003:** Defining Cultural Safety, Cultural Competence, and Similar Concepts (Baba 2013) [34].

Concept	Definition
Cultural Awareness	The acknowledgement and understanding of cultural differences by focusing on the “other” and the “other culture”. Does not consider the political or social-economic influences on cultural difference and does not require an individual to reflect on his/her own cultural perspectives.
Cultural Sensitivity	Recognizes the need to respect cultural differences. Involves exhibiting behaviours that are considered polite and respectful by the persons of other cultures. Focuses on the “other” and the “other culture” and does not require an individual to reflect on his/her own cultural perspectives.
Cultural Competence	Skills and behaviours that help a practitioner provide quality care to diverse populations. While cultural competence can build upon self-awareness, it is limited by reducing culture to a set of skills for practitioners to master and over-emphasizes cultural differences as the course of conflict between healthcare providers and diverse populations.
Cultural Safety	Cultural safety within an Indigenous context means that the educator, practitioner, or professional, whether Indigenous or not, can communicate competently with a patient in that patient’s social, political, linguistic, economic, and spiritual realm. Moves beyond cultural sensitivity to analyzing power imbalances, institutional discrimination, colonization, and colonial relationships as they apply to healthcare.

## Data Availability

A Preprint of this manuscript is available online via BioRxiv (www.borxiv.org, accepted on 7 October 2022). https://doi.org/10.1101/2022.10.06.511097.

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
