# Peer review of "Indigenous Cultural Safety Training for Applied Health, Social Work, and Education Professionals: A PRISMA Scoping Review"

_ijerph, 2023, doi:10.3390/ijerph20065217_

Round 1

Reviewer 1 Report

Comments to the authors

This is an interesting manuscript. The article needs organization and some changes. See other comments below.

Abstract

1. The abstract should include the design. Check it, please.

2. Keywords: include “metasynthesis”.

Introduction and Background:

1.- Data published in 1998,1999,2000.. Please, searches must be up to date. Check the information sources. There are some works that have been updated and show more current data. I suggest including more recent research that examines the problem.

Methods

Need to give more detail on inclusion/exclusion criteria.

Academic literature between 1996-2020 were included. Why 2020 and not include until 2022? Why have not been included prior to 1996?

More detail must be given on the process of the final selection of articles using PRISMA. How was this process done?

Results

The results section is very synthetic. Please, expose all the significant results obtained.

Discussion

Some reorganization is required. The discussion is correct but somewhat disordered. I suggest writing the discussion in more order, giving the information according to the stated objective and results.

Strengths and limitations

Very honest.

Conclusion

I suggest focusing the conclusion on the main ideas that the study brings to the world. Be sure to mention exactly what is your study contribution.

Author Response

Point 1: The reviewer suggested that we include the study design in the abstract.

Response 1: The term 'Scoping' has been added to the term "review" in the methods section of the abstract. Thank you for catching this.

Point 2: The reviewer also suggested to include the term “meta-synthesis” into the selection of keywords.

Response 2: We also greatly appreciate this suggestion and the term “meta-synthesis” has been added to the keywords.

Point 3: The reviewer referred to references in the background section of the paper that dated back to 1999-2000, and suggested that we use more recent sources so that our searches are up to date.

Response 3: Thank you for this comment. We think that the dates that you mentioned are from the background section of the paper and refer specifically to large population based studies of Indigenous peoples' experiences of racism in Canada. These are the only studies of their kind undertaken in Canada, and we felt it important to include them in the background section to highlight the urgent need for cultural safety training. As for the articles specifically included in the scoping review, they date back to 1996 because this is the year the concept of cultural safety was developed and published by a Maori nurse. As identified in the findings of our paper, over two-thirds of the papers included in the review were published between 2011-2020, with far fewer published in the previous two decades dating back to 1996. I hope that helps to clarify the article publication years we selected for the review and why they were selected. 

Point 4: More details were requested about the inclusion/exclusion criteria.

Response 4: Thank you for catching this oversight. While the inclusion criteria were complete, in section 3.2 (eligibility criteria) of the methods section, we have added the exclusion criteria, which include: articles published prior to 1996; articles published in countries other than New Zealand, Australia, United States or Canada; articles focused on fields other than health, social work, or education; and research published as non-academic literature or in languages other than English.

Point 5: The reviewer questioned why the review did not include articles beyond 2020 (up to and including 2022), and prior to 1996.

Response 5: Thank you for this question. In section 3.2 of the methods section, we outlined that the review start year was 1996 because this was the year the term cultural safety was first coined, by a Maori nurse in New Zealand. Moreover, we added to the limitation section that that this review is somewhat dated, as the searchers were undertaken in mid-2020, and in the early stages of the Covid-19 pandemic, and with no dedicated research funding. Meanwhile, new priorities to understand and address Indigenous peoples’ experiences with the Covid-19 became competing priorities with this study, and at a time when there was increased demand for Indigenous CST. As a result, our team used the preliminary findings from this review to prioritize the development of Oshki M’naadendimowin, a 36-hour Indigenous cultural safety training program for faculty members, staff and students at the Faculties of Medicine, Nursing, Social Work and Public Health at the University of Toronto. For these reasons, combined with the large number of articles and data categories included in this review, we were not able to update the article searches to 2023. 

Point 6: More detail must be given on the process of the final selection of articles using PRISMA. 

Response 6: we have included extra details in section 4.2: (Results) Selection of sources of evidence (lines 265-284). These include the following: that the reviews were conducted in may 2020, and performed in Covidence, which is an online research platform specifically developed for screening articles for review, so it make this process seamless, including by identifying duplicates studies, among other features. We've also added details about applying the exclusion criteria, and that any disagreement between the two reviewers on applying these criteria was resolved through discussion. PRISMA wasn't used to select studies, they are broad guidelines for conducting Scoping Review, and which we followed for this study.

Point 7: The suggestion was made to consider removing or reorganizing tables 1-5 in methods section. As data is repeated in text. 

Response 7: Thank you for this suggestion. We have removed Tables 1-3, which repeated the same search terms for four different databases. Instead we have harmonized the search terms across the three tables and included them into one easy-to-read table, removing the search strings to make it more reader friendly. (Table 1: Search Strategy, line 152 of the revised paper). We also deleted table 4 (scoping search result numbers by database) as most of this information is in the prisma diagram. And we have included Table 5 (data themes and focus) as  Appendix 1. 

 Point 8: The results section is very synthetic. Please, expose all the significant results obtained. 

Response 8: thank you for your comment. There are a couple of reasons why we chose to present the data in this way. First, the review sought to synthesize the state of knowledge on ICS Training in the applied fields chosen, so this was our objective from the beginning. Second, the large number of articles we included in the review (n=134), along the substantial number of data points we extracted, and without any dedicated funding for this study, was why we chose to synthesize the literature. Third, our team wanted to use this data to develop an ICS Training intervention for staff, students and Faculty at the University of Toronto, so our priority was to publish of synthesized data. Since this review however, we have used the findings from this review to do secondary research and have writing additional papers on ICS training for specific fields. We have already published one for education, and we are currently finishing a paper for Social Work. We hope this data base can be used by other researchers as well for secondary analysis, as it is publicly available on BioRxiv (the link is in the paper).

 Point 9: You suggested to reorganize the discussion section, giving the information according to stated objective and result.

 Response 9: We greatly appreciated this suggestion and to clearly state how it addressed the paper objective. Thank you for this. We hope you will find that the discussion now clearly presents the key findings and contributions of the paper and identifies areas for future research.

Point 10: You suggested that we focus on the main ideas that the study brings to the world in the conclusion section. And to be sure we mention exactly what is your study contribution.

Response 10: we have reduced this section and more clearly outlined the key contribution of this work as a synthesis of the substantive work undertaken over the last three decades in Can, US, NZ and AUS, on ICS Training in the applied fields of health, social work and education (please see p21).

Reviewer 2 Report

Dear authors,

thank you for the opportunity of reviewing this manuscript about the Indigenous cultural safety training for applied health, social work and education professionals. In my opinion, this manuscript is interesting. However, there are some points to be considered before publication:

ABSTRACT

The abstract needs revisions to better present important findings. And please remove the references from the abstract.

INTRODUCTION

In order to reduce the introduction and overall manuscript, consider removing the text line 60-67.

METHODS

Methods are well described. However, they take up a lot of space in the manuscript. Please consider the possibility of removing or reorganizing all tables from 1 to 5, since the text clearly presented a lot of it.

RESULTS

Most of the result text is occupied by reference numbers, please rearrange because table 6 shows everything anyway.

CONCLUSION

Please focus on your main aim when concluding your findings. The current one is too extensive.

Author Response

Point 1: Revise Abstract findings section to better present important findings. Remove numbers.

Response 2: Thank you for these suggestions. We have removed the numbers/references from the abstract, and we have revised the abstract to include the key findings. Namely, "CST programs have grown significantly in in the health, social work and education fields in the last three decades, and they vary significantly in their objectives, modalities, timelines, and how they are evaluated. The involvement of Indigenous peoples in CST programs is common, but their roles are rarely specified. Indigenous groups must be intentionally and meaningfully engaged throughout the entire duration of research and practice. Cultural safety and various related concepts should be careful considered and applied for the relevant context." This change reflects the changes we've also made to the Discussion and conclusion sections.

Point 2: To reduce the introduction and overall manuscript, consider removing the text line 60-67. 

Response 2: Thank you for this suggestion. We have removed lines 60-67 from the introduction.

Point 3: Most of the result text is occupied by reference numbers, please rearrange because table 6 shows everything anyway. 

Response 3: Thanks for this suggestion. We removed the citations/numbers from the results section, and have instead placed them with the findings in table format as Appendices. We have done this because citing each finding is a requirement of the PRISM Scoping Review guidelines, which we have followed. See Appendix 3 (Citations for General Characteristics of Sources of Evidence), Appendix 4 (Citations for Indigenous Cultural Safety Training Interventions) and Appendix 5 (Citations for Indigenous Cultural Safety Training Intervention Evaluations).

Point 4: Please focus on your main aim when concluding your findings. The current one is too extensive.

Response 4: thank you for this suggestion. We have reduced this section and more clearly outlined the key contribution of this work as a synthesis of the substantive work undertaken over the last three decades in Can, US, NZ and AUS, on ICS Training in the applied fields of health, social work and education (please see p43).

Reviewer 3 Report

Dear Authors,

Indigenous Cultural Safety Training for Applied Health, Social Work and Education Professionals: A PRISMA Scoping Review

The authors have conducted a scoping review to understand existing evidence on how such training programs are conceptualized and developed, as well as how they are implemented in practice and evaluated for impact. The paper addresses a very relevant issue and is methodologically clear. However, the presenting style and language can be improved, and there are a lot of unnecessary detailing and repetitions throughout the manuscript, significantly reducing the overall quality.  I strongly suggest that the authors refer to a few already published scoping reviews to improve the presentation of the methodology and the findings. Here are a few specific suggestions to improve the manuscript.

·         Line 122 – include the exclusion criteria

·         Why wasn’t Scopus used to collect information? Since the objectives clearly mention social work, it is advised to include a social science database.

·         Please use a consistent style for in-text citations – line 131

·         Please avoid repeating information –

o   For instance, You can describe the search strategy used – need not separately include search strings used for each database (refer to table 1 https://www.ncbi.nlm.nih.gov/pmc/articles/PMC9617183/ to present search strategy)

o   Similarly, lines 151 – 160 repeats eligibility criteria which can be avoided

o   Delete lines 259 - 264

·         Please list the major components of the data charting tool – what information was gathered from the selected articles (the major heads)?

·         There is much unnecessary detailing within the manuscript which can be avoided –

o   Table 4 doesn’t add anything new. Please consider deleting it. You can merge the information from table 4 into the PRISMA Diagram.

o   Table 5 can be provided as a supplementary file.

o   Lines 226 – 237 – what is the relevance of this paragraph to the current review?

·         Section 4.2 – General characteristics – please avoid the number citations – I see nothing but the numbers in the whole paragraph!!

·         Table 6 adds nothing new – it’s already described in section 4.2.

·         Please refer to an already published scoping review to see how results are presented – please include two tables – characteristics of studies and a summary of findings as you would commonly see in a scoping review – refer to https://www.ncbi.nlm.nih.gov/pmc/articles/PMC9617183/ or any other published review for an overview of results.

·         Table 9 is missing; also, a total of nine tables in a manuscript is a little too much!!

·         Are the definitions given in table 10 the work of the authors, or are they adapted from elsewhere? – if it is taken from somewhere, please give references.

·         Please include a quality assessment of the selected studies – please use JBI or CASP framework to assess the quality and present the results.

·         Please summarize the conclusion into a paragraph or two.

Overall, the manuscript requires significant reworking – especially for the methods and results section before it can be considered for publication.

Author Response

Point 1: The reviewer suggests to give more detail on inclusion/exclusion criteria.

Response 1: Thank you for catching this oversight. While the inclusion criteria were complete, in section 3.2 (eligibility criteria) of the methods section, we have added the exclusion criteria, which include: articles published prior to 1996; articles published in countries other than New Zealand, Australia, United States or Canada; articles focused on fields other than health, social work, or education; and research published as non-academic literature or in languages other than English.

Point 2: The reviewer asks why wasn’t Scopus used to collect information? Since the objectives clearly mention social work, it is advised to include a social science database.

Response 2: Thanks for your question. In consultation with the librarian at the University of Toronto, the databases we used ASSIA (Applied Social Sciences Index and Abstracts) to cover social work. We also used ERIC (Education Resource Information Centre) to cover education. The other databases include MEDLINE, EMBASE, and CINAL, which together covers most of the applied health professionals. So the broad fields we've included were covered by these databases.

Point 3: The reviewer suggests to please use a consistent style for in-text citations 

Response 3: Thank you for catching this, we have changed the one odd citation on line 131 so that its consistent with the Vancouver style we chose for the paper.

Point 4: The reviewer suggests to just describe the search strategy used – need not separately include search strings used for each database.

Response 4: Thank you for this wonderful suggestion and the example table you provided. We have removed the three tables with complex search strategies and strings, and replaced them with a harmonized table including search terms used for all four databases. 

Point 5: General characteristics – please avoid the number citations – I see nothing but the numbers in the whole paragraph!!

Response 5: Thanks for this suggestion. We removed the citations/numbers from the results section, and have instead placed them with the findings in table format as Appendices. We wanted to provide citations for each finding as this is a requirement of the PRISM Scoping Review guidelines, which we have followed. See Appendix 3 (Citations for General Characteristics of Sources of Evidence), Appendix 4 (Citations for Indigenous Cultural Safety Training Interventions) and Appendix 5 (Citations for Indigenous Cultural Safety Training Intervention Evaluations).

Point 6: The reviewer suggests Table 9 is missing, and a total of nine tables in a manuscript is a little too much

Response 6: We have reduced the number of tables in the paper from 9 to 3. Table 1 presents the harmonized search strategy across 4 databases (Methods). Table 2 presents the General Characteristics of Sources of Evidence (Results). Table 3 presents the definitions of various cultural safety and related concepts (Discussion).

Point 9: Are the definitions given in table 10 the work of the authors, or are they adapted from elsewhere? – if it is taken from somewhere, please give references.  

Response 9: The source of information from table 3 (formerly table 10) was mentioned in the paragraph above it, but I have also included the source in the table header. Thanks for pointing this out.

 Point 10: Please include a quality assessment of the selected studies – please use JBI or CASP framework to assess the quality and present the results. 

Response 10: We have followed the PRISMA guidelines for Scoping Reviews, which views quality appraisals as optional. We chose not to appraise the studies as the aim of the paper was to broadly broadly synthesize the academic literature on how CST programs are developed, implemented, and evaluated in the applied health, social work and education fields in Canada, United States, Australia, and New Zealand. Given the lack of dedicated funding for this study, and the large number of articles (n=134) and data points included, we decided not to conduct an appraisal.

 Point 10: Please summarize the conclusion into a paragraph or two. 

Response 10: Thank you for this suggestion, we have reduced the conclusion to two paragraphs.

Reviewer 4 Report

The authors have done an excellent job of collecting, reviewing, and summarizing the studies to answer the research questions posed.   The methodology used is explained clearly, with strengths and limitations identified.   The discussion and recommendations make sense and reflect results uncovered through the analysis. 

The narrative presentation of the results (with citing the number of the study) looks a bit busy.   For the most significant findings, a  matrix table indicating what criteria each study meets may enhance the readability and make visible overall pattern of this literature.  For example:

Study (Author(s)

Criteria A

Criteria B

Criteria C

Criteria D

Criteria E

Criteria F

Criteria…

Study A

X

X

X

Study B

X

X

X

X

Study C

X

X

X

x

X

Author Response

Point 1: The narrative presentation of the results (with citing the number of the study) looks a bit busy. For the most significant findings, a  matrix table indicating what criteria each study meets may enhance the readability and make visible overall pattern of this literature. 

Response 1: Thank you for kind comments and your suggestion to improve the readability of the result. The other reviewers agreed with you.  We have addressed this problem by removing the citations (i.e. numbers) from the results section, and have instead placed them with the findings in table format as Appendices (because citing each finding is a requirement of the PRISM Scoping Review guidelines, which we have followed). See Appendix 3 (Citations for General Characteristics of Sources of Evidence), Appendix 4 (Citations for Indigenous Cultural Safety Training Interventions) and Appendix 5 (Citations for Indigenous Cultural Safety Training Intervention Evaluations). We you hope you will find that this address the readability problem of the results section. As we included 134 studies, using a matrix would have been a bit challenging.

Round 2

Reviewer 1 Report

The paper has improved considerably. All suggestions have been incorporated. Congratulations to the research team.

Reviewer 3 Report

I appreciate the authors for the corrections made to the manuscript - it has come out well. All the best for all your future endeavors.